# Pure Sinusoidal Output Single-Phase Current-Source Inverter with Minimized Switching Losses and Reduced Output Filter Size

**Eka Rakhman Priandana** [1,2,*] and **Toshihiko Noguchi** [3,*]

1 Department of Environment and Energy System, Graduate School of Science and Technology, Shizuoka University, Hamamatsu 432-8561, Japan
2 National Laboratory for Energy Conversion Technology (B2TKE) of Agency for the Assessment and Application of Technology (BPPT), Tangerang Selatan 15314, Indonesia
3 Department of Engineering, Graduate School of Integrated Science and Technology, Shizuoka University, Hamamatsu 432-8561, Japan
* Correspondence: aan.woodz@gmail.com (E.R.P.); noguchi.toshihiko@shizuoka.ac.jp (T.N.); Tel.: +81-70-4432-3433 (E.R.P.); +81-53-478-1128 (T.N.)

**Abstract:** This paper proposes a novel single-phase current-source inverter that generates a pure sinusoidal waveform with minimized switching losses and using a small-size output filter capacitor. The proposed method is investigated by incorporating a conventional multilevel current-source inverter with a linear amplifier. The conventional multilevel technique uses fundamental switching frequency instead of using high-switching frequency modulation for the H-bridge circuit. The linear amplifier such as class-A or class-D types has a function to reform the staircase waveform generated by the multilevel inverter into a pure sinusoidal by using superimposition technique. As a result, pure sinusoidal output current is generated with a small ripple and the system only requires a small output filter capacitor for smoothing the waveform. Based on the simulation and experimental results, the proposed system presents not only the optimal configuration, but also an option as to whether to obtain excellent power efficiency or very low output harmonic. Implications of the results and future research directions are also presented.

**Keywords:** current-source; multilevel inverter; linear amplifier; superimposition

## 1. Introduction

Direct current (DC) to alternating current (AC) converters, or inverters, are basically operated by means of a switching action to optimize their power conversion efficiency. In order to minimize the output noise or harmonics caused by switching modulation, a passive L-C filter is indispensable [1]. By contrast, linear power amplifiers such as class-A, class-AB, or class-D can generate a pure sinusoidal waveform with very low noise. However, poor power conversion efficiency is their main issue, so that they are not suitable for power inverters [2–4]. The primary goal when constructing a power inverter is generating pure sinusoidal waveform with very low output harmonics without forfeiting the power efficiency [1]. Therefore, researchers and engineers have adopted several approaches to reach the requirement by either using high-switching frequency modulation, enlarging the output L-C filter size, or employing multilevel technique. The latter features generate voltage or current with extremely low distortion and low *dv/dt* or *di/dt*. The multilevel technique also draws input current with very low distortion. The most important is that it can operate with low-switching frequency so that the switching losses can be minimized [5–10].

Generally, power inverters are classified into two topologies, voltage-source inverters (VSIs) and dual circuit, current-source inverters (CSIs) [11]. The CSIs are well-known for having high capability to deliver very high current for inductive load, having robustness against the grid voltage fluctuation, and having low *dv/dt* or *di/dt* transient behavior [12–15]. However, the key problems with this topology are low efficiency caused by high conduction losses and low power density that is caused by inductor-based energy buffer that makes them less being developed and applied [14,16,17].

A number of works have shown that these efficiency and power density problems can be overcome by applying the multilevel technique into CSIs. So far, however, there has been little discussion about multilevel CSI (MCSI). The conventional method for generating a multilevel current waveform is by paralleling some H-bridge CSIs. This topology is a dual circuit of cascaded multilevel VSI (MVSI) [18,19]. However, several isolated current sources are needed for this topology. In order to reduce the number of isolated current sources, several approaches were introduced. The first one is a nested structure as an advanced H-bridge technology. The second one is a dual circuit of diode-clamped MVSI. The third one is an inductor multi-rating as a dual circuit of flying-capacitor MVSI [19–22]. However, these topologies employed enormous inductors as energy buffers so that their power density was very low.

Recent developments in MCSI become revolutionary after several novel topologies were proposed by Suroso and Noguchi. They invented the fish-bone structure of MCSI as a dual circuit of 3-level diode-clamped half-bridge MVSI and then they expanded it for a higher number of levels. This topology became an alternative of the conventional H-bridge type MCSI and its derivative such as paralleled and nested structures [1,12,23–25]. They also developed an inductor-cell based MCSI technology as a dual circuit of capacitor cell-based MVSI [1,12,15,26] and a DC current module technology [1,12,13] for generating multilevel current waveform by using single H-bridge CSI. However, they all were operated with high-switching frequency modulation in order to reduce the output L-C filter size. Unfortunately, this effort results in problems related to the switching losses. One approach to solving this problem involves the use of linear power amplifiers. It is of interest to know whether the linear power amplifier application in power inverters still holds to be true and reasonable.

The objective of the present work paper is to investigate the linear power amplifier application could improve the MCSI system performances in power efficiency, power quality, and power density. As described in Figure 1, the proposed concept is composed of several constant current sources, several switching modules, a linear current generator, a polarity alternator represented by a CSI H-bridge circuit, and several diodes. Constant current sources are the main power source. Switching modules control the direction of each current generated by constant current sources in order to generate staircase current waveform. The linear current generator produces linear current to compensate the staircase current into pure sinusoidal waveform. The CSI H-bridge circuit inverts the DC current into AC. As a CSI characteristic, several diodes are required in the circuit to prevent the reverse current. Due to combination between the staircase current waveform and the linear current waveform generations, this proposed MCSI is simply called a hybrid MCSI.

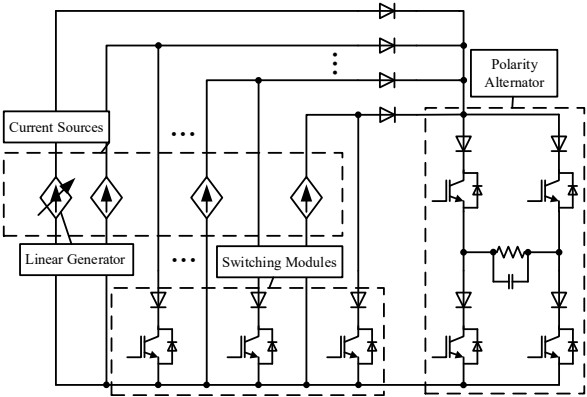

**Figure 1.** Generalized concept of hybrid multilevel current-source inverters (MCSI).

The contributions of this work are presented as follows:

1.　No high-switching frequency modulation is used for current inversion processing.

　　The proposed method applies two times fundamental frequency of the output for controlling the current direction by switching modules in order to generate staircase current waveform. In other words, without any high-switching frequency modulation. Therefore, the switching losses in the current inversion process or in the H-bridge part is extremely low. Nevertheless, the high-switching frequency modulation is still needed for generating constant current and linear current sources.

2.　No dedicated linear amplifier is used for generating pure sinusoidal output.

　　The amount of energy for constructing pure sinusoidal output current is still dominantly supplied by constant current sources through a staircase current waveform. The linear amplifier produces only a very small amount of amperage to compensate and reshape the staircase current waveform into pure sinusoidal. Hence, the power losses wasted by linear amplifier circuit can be minimized.

3.　No large output L-C filter utilization.

　　The size of the utilized output L-C filter affects the power density [27]. Due to linear amplifier utilization, the output total harmonic distortion (THD) is always being kept low in order to comply with IEEE-1547, IEEE-929, and EN-61000-3-2 standards [26]. Therefore, the proposed system only needs small output capacitor that is sufficient to eliminate the current ripples. As a result, the power quality and the power density of the proposed system are slightly improved.

4.　No isolated multiple power sources utilization.

　　Figure 1 describes the proposed MCSI concept employs several isolated current sources. Moreover, the research finding by author [1] also points towards that the proposed concept is consistent with literature. However, the implementation is totally different. The depicted current sources actually can be represented by only using one power source. The detailed implementation will be elaborated in the next section.

This paper is organized as follows: first, a brief introduction and significances of this research are given. Next, the operation principle and circuit configuration of the proposed system are described in detail. In order to verify the proper operation of the proposed system, several simulation and experimental results are presented, respectively. Finally, the conclusion and future works are addressed at the end of this paper.

## 2. Operation Principle, Circuit Configuration, Switching Strategy and Control System

In this section, the basic operation principle and the circuit configuration of the proposed system are described. In addition, the proposed control system as well as the switching strategy and a theoretical calculation for conduction and switching losses of the proposed system are also elaborated. Several studies have attempted to explain that the linear power amplifier for audio applications can also be utilized in power converter applications [1,28]. However, author [1] has problems in representing single power source usage and low-switching frequency applications, meanwhile Author [28] has a problem in representing non-audio applications. An as yet unsolved question is whether the linear amplifier application in multilevel CSI could generate pure sinusoidal output with using single power source and without using high-switching frequency modulation as well as large output L-C filter. There is still considerable controversy surrounding the linear amplifier application in power electronics. Additional studies to understand more completely the key tenets of linear amplifier application in power electronics are required.

### 2.1. Operation Principle

As shown in Figure 1, the proposed hybrid MCSI consists of several constant current sources and switching modules, a linear current source, and a polarity inverter that is represented by a CSI H-bridge circuit. The operation principle of the proposed system is illustrated simply in Figure 2.

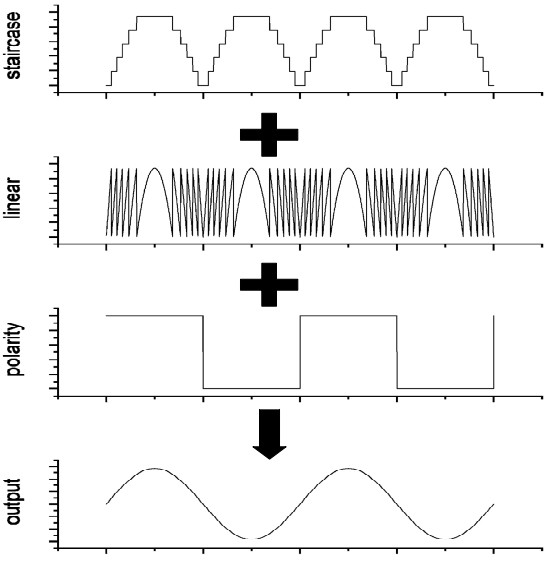

**Figure 2.** Operation principle.

As explained beforehand, the proposed system applies several current sources as the main power source. The number of required current sources that corresponds to the number of levels is expressed by:

$$M = \frac{N_{LEVEL} - 1}{2} \tag{1}$$

where $M$ is the number of current sources and $N_{LEVEL}$ is the number of levels. The number of constant current sources that equals to the number of switching modules is written as:

$$CCS = M - 1 \tag{2}$$

where $CCS$ is the number of utilized constant current sources or the number of switching modules in the system. This $CCS$ also represents the number of steps or stages that exposed by the generated staircase current waveform.

Figure 2 describes the basic operation principle of 13-level hybrid MCSI. By using Equation (1), the number of required current sources is six. According to Equation (2), five current sources are programmed to generate constant current whose amperage is determined by the main controller of the system. While the remaining current source is for the programmable linear current generator. Each constant current source generates equal DC current which is the direction is controlled by switching modules in order to generate staircase waveform on the DC link current. While the linear current generator compensates the DC link current to result the DC full-wave current waveform. The CSI H-bridge circuit changes the polarity of the load current every time a zero-cross of sinusoidal reference is detected.

### 2.2. Direct Current (DC) Module

It is inefficient to utilize several dedicated current sources. The system can be improved if only using one power source. To make this happen, the DC current source is constructed from the buck chopper circuit. The buck chopper converts the voltage into a current [12,13,25]. In order to simplify

the circuit, a switching module is then combined with the buck chopper circuit. This combined circuit is simply called DC current module (DCM) as depicted in Figure 3. The operation principle of the DCM is explained in Table 1.

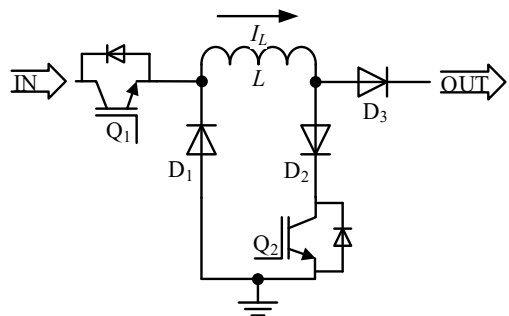

**Figure 3.** Direct current module (DCM) circuit.

**Table 1.** DC current module (DCM) operation principle.

| Q1 | Q2 | Output |
|---|---|---|
| 0 | X [1] | 0 |
| Switching | 0 | $I_L$ |
| X [1] | 1 | 0 |

[1] X = do not care.

The number of DCMs equals to the number of required constant current sources, therefore it is also denoted as CCS. A DCM consists of switch Q1, inductor L, free-wheeling diode D1, blocking diodes D2, blocking diode D3, and switch Q2. The amount of energy that generated by switching the Q1 is preserved in L and then released as DC current $I_L$. In order to reduce the L size, a high-switching frequency modulation is applied for Q1. D1 is used to keep continuous conduction of the chopper part by circulating the load current. In order to maintain $I_L$ stability, a close-loop controller is applied for this circuit. The switching module part directs $I_L$ whether to deliver it to the load or circulate back to the source. When Q2 is turned off, $I_L$ flows to the load before flowing back to the source through D1. On the other hand, when Q2 is turned on, $I_L$ flows directly back to the source so that no current supplies the load. The switching module part is working based on the comparison between the absolute value of the sinusoidal reference with the determined current limit of each DCM.

*2.3. Circuit Configuration*

The final proposed hybrid MCSI system is shown in Figure 4 which includes a power source, several DC current modules, a linear current generator, and a CSI H-bridge circuit.

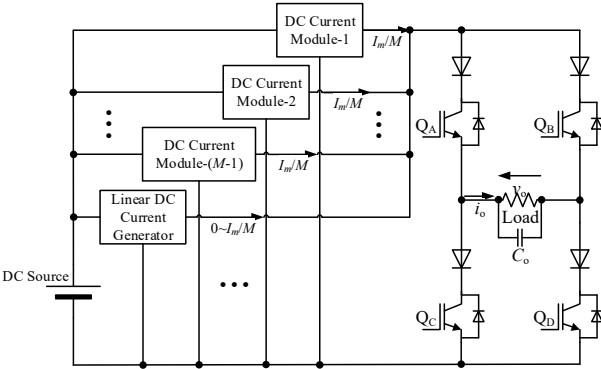

**Figure 4.** Proposed design.

In Figure 4, the AC current $i_O$ supplies the load so that giving the load a potential difference of $v_O$. In order to keep the output voltage stability, $i_O$ must be maintained at:

$$i_O = \frac{v_O}{Z} \tag{3}$$

If the load is assumed as pure resistive $R$, therefore the output impedance $Z$ is:

$$\frac{1}{Z} = \frac{1}{R_{LOAD}} + j\omega C_O \tag{4}$$

Because $i_O$ is sinusoidal waveform, it has a peak or maximum value that equals:

$$I_m = \sqrt{2}i_O \tag{5}$$

This $I_m$ determines the required DC link current. Because the $I_m$ robustness is the key of $v_O$ stability, therefore the current generated by each current source is limited to:

$$I_{CS} = \frac{I_m}{M} \tag{6}$$

For instance, if the desired MCSI level is 13-level, then $I_{CS}$ is limited to $I_m/6$ amperes. Five DCMs are set to generate constant current of $I_m/6$ amperes, meanwhile the linear current generator is programmed to generate linear current from 0 to $I_m/6$ amperes.

The output filter capacitor $C_O$ is added in order to eliminate the ripple current that accumulated from each current source as a consequence of using the buck chopper circuit. In order to reverse the current polarity of the load, the operation of $Q_A$ is paired with $Q_D$ and the operation of $Q_B$ is paired with $Q_C$. When a zero-cross of the sinusoidal reference is detected, prior to changing the polarity, switches $Q_A$–$Q_D$ are turned on for a few microseconds to make sure the DC current does not lose its current path. This is well-known as overlap time that always required by the CSI system to prevent any damages.

### 2.4. Staircase Current Waveform Generation

The staircase current generator is working based on the comparison between the reference with the current limit of each stage. The reference of this generator is obtained from the absolute value of the sinusoidal waveform. In other words, the reference is DC full-wave waveform with unity maximum value. Figure 5 explains how the staircase current waveform is generated.

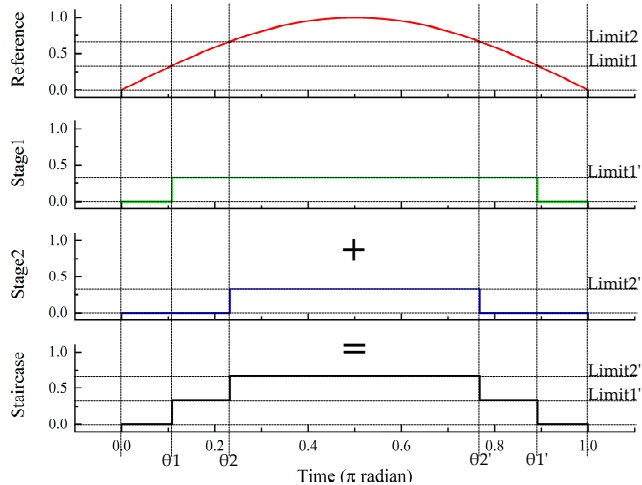

**Figure 5.** Staircase current waveform generation.

Figure 5 shows an example of staircase current waveform generation of 7-level hybrid MCSI. Limit1 and Limit2 are determined by the number of required current sources *M*. Stage1 is the first square-wave current generator and Stage2 is the second. Limit1 is the first comparison level that equals 1/*M* which its comparator commands to turn on and off the Stage1. Limit2 is the second or the last comparison level that equals 2/*M* or (*M*–1)/*M* which its comparator commands to turn on and off the Stage2. When the reference goes up until it reaches the Limit1, Stage1 is turned on at θ1 with steady amplitude Limit1'. The Stage1 will be turned off at θ1' when the reference has been lower than the Limit1. Similarly, when the reference continues going up until reaches the Limit2, Stage2 is turned on at θ2 with steady amplitude at Limit2'. The Stage2 will be turned off at θ2' after the reference has been lower than the Limit2. The staircase current waveform is resulting from the summation of Stage1 and Stage2 outputs.

The square-wave current waveform of Stage1 and Stage2 has an average value that is expressed by:

$$I_{SQ\_AVG} = I_{CS}D \tag{7}$$

and

$$D = \frac{T_{ON}}{T_{ON} + T_{OFF}} \tag{8}$$

where $I_{CS}$ is the current generated by the corresponding current source of the stage and *D* is the duty cycle that equals to the turn-on period divided by the total period-length. In Figure 5, θ1 and θ2 can be found by:

$$\theta_n = \arcsin\frac{n}{M} \tag{9}$$

where *n* is the sequence number of the current limits. Therefore, the duty cycle of the square-wave can be calculated as follows:

$$T_{ONn} = \pi - 2T_{OFFn} = \pi - 2\theta_n \tag{10}$$

$$D_n = 1 - \frac{2\theta_n}{\pi} \tag{11}$$

By synthesizing Equations (6)–(11), the average value of the square-wave current waveform of Stage-n can be estimated by:

$$I_{SQn\_AVG} = \frac{I_m}{\pi M}\left(\pi - 2\arcsin\frac{n}{M}\right) \tag{12}$$

According to Kirchoff's current law (KCL), the total output currents equal to total input currents. Therefore,

$$I_{STAIRCASE} = I_{SQ1\_AVG} + I_{SQ2\_AVG} + \ldots + I_{SQn\_AVG} \tag{13}$$

By generalizing Equation (12) to Equation (13), the average value of the staircase current ($I_{STAIRCASE}$) waveform can be written as:

$$I_{STAIRCASE} = \frac{I_m}{\pi M}\sum_{n=1}^{M-1}\left(\pi - 2\arcsin\frac{n}{M}\right) \tag{14}$$

The proposed method conveys a new technique for generating pure sinusoidal output, i.e., applying a linear current compensator to eliminate harmonics. There have been numerous studies to investigate harmonic reduction of the staircase waveform by using complex mathematics [29,30]. However, previous research never discussed other methods than just using a capacitor as the final output harmonic filter. In this proposed approach, simple mathematical equations are used.

*2.5. Linear Current Waveform Generation*

The linear current compensator is used for reforming the staircase current waveform into pure sinusoidal waveform by means of superimposing it. Due to Kirchoff's current law, the linear current

compensator waveform is obtained by subtracting the DC full-wave reference with the generated staircase current waveform as described in Figure 6. The average value of DC full-wave current reference is expressed by:

$$I_{AVG} = \frac{2I_m}{\pi} \tag{15}$$

where Figure 6 explains that:

$$I_{AVG} = I_{STAIRCASE} + I_{LINEAR} \tag{16}$$

Therefore, the average of linear current ($I_{LINEAR}$) waveform can be estimated as:

$$I_{LINEAR} = \frac{I_m}{\pi}\left[2 - \frac{1}{M}\sum_{n=1}^{M-1}\left(\pi - 2\arcsin\frac{n}{M}\right)\right] \tag{17}$$

In order to generate the linear current waveform, a linear power amplifier is required. For the proposed circuit shown in Figure 4, there are two classes of linear amplifier can be used. They are class-A and class-D amplifiers. Due to dual supply requirement, class-B/AB amplifier is not suitable for this kind of proposed hybrid MCSI topology.

### 2.5.1. Class-A Amplifier-Based Linear Current Generator

The class-A amplifier is well-known the most linear among all linear counterparts. However, its power efficiency is very challenging [2,4]. There are three configurations of transistor-based class-A amplifier circuit: common-emitter, common-base, and common-collector. A common-emitter configuration is the default circuit for a general-purpose amplifier. It does not only amplify the current, but also the voltage. Common-base configuration has a characteristic to amplify the voltage only. Common-emitter configuration also known as emitter-follower only amplifies the current.

In case of linear current generation for the proposed hybrid MCSI, a class-A amplifier with common-collector or emitter-follower configuration is chosen due to voltage unity amplification. The complete circuit of class-A amplifier-based linear current generator is shown in Figure 7.

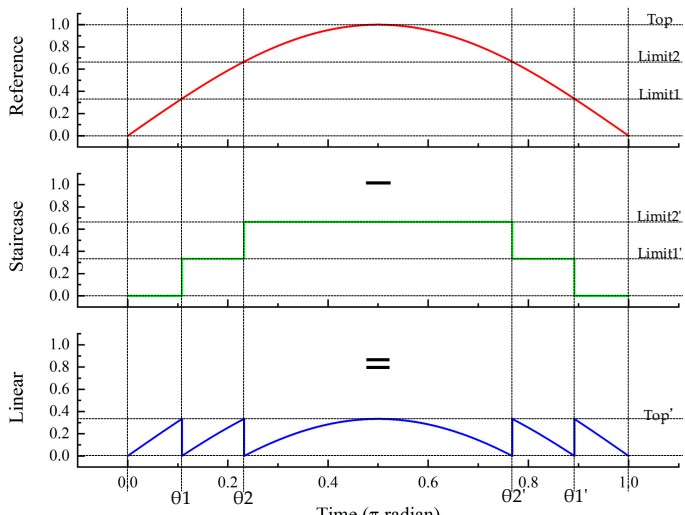

**Figure 6.** Linear current waveform generation.

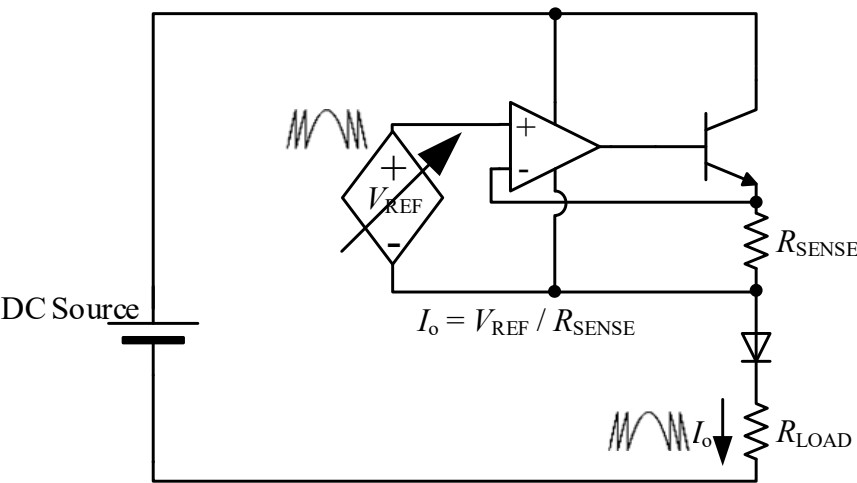

**Figure 7.** Class-A amplifier-based linear current generator.

In Figure 7, the op-amp is used for maintaining the sense-resistor voltage, meanwhile the power transistor provides sufficient current to the load due to the op-amp limited output current. In order to prevent the reverse current, a blocking diode is inserted prior to the load. In the class-A amplifier-based system, an analog voltage reference is needed for controlling the output current. Hence, the digital-based system controller must employ the digital-to-analog converter (DAC) in order to drive the amplifier.

### 2.5.2. Class-D Amplifier-Based Linear Current Generator

Class-D amplifiers are increasingly ubiquitous as the audio power amplifier in audio devices due to their significantly higher power-efficiency compared to their linear counterparts [3]. They operate by rapidly switching back and forth between the supply rails, being fed by a modulator using pulse width, pulse density, or related techniques to encode the linear input into a pulse train. The linear waveform escapes through a filter into the load while the high frequency pulses are blocked.

Figure 8 describes a class-D amplifier-based system for generating linear current waveform. The input is compared with the sensed output current in order to identify the error. This error is amplified with proportional-integral controller and then modulated by high frequency sawtooth carrier waveform. The generated pulse train then drives the switch. The inductor $L$ is used for smoothing the pulse train current and blocking high frequency pulses. As a result, a linear current waveform is generated at the output which is the replica of its input.

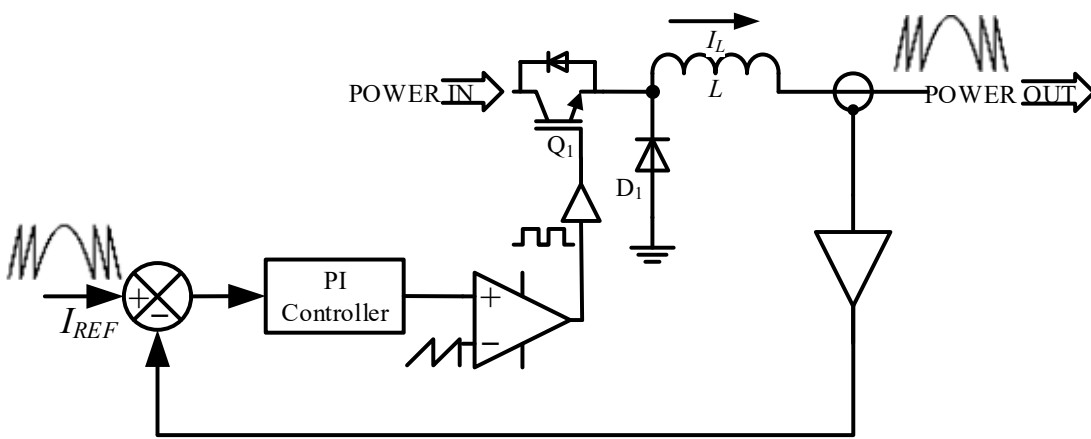

**Figure 8.** Class-D amplifier-based linear current generator.

### 2.6. Control System

In order to maintain the output robustness, the load voltage and line frequency must be controlled. The load voltage is controlled by controlling the DCM output current, while the line frequency is controlled by using phase-locked loop (PLL). Figure 9 shows the general scheme of the proposed hybrid MCSI control system.

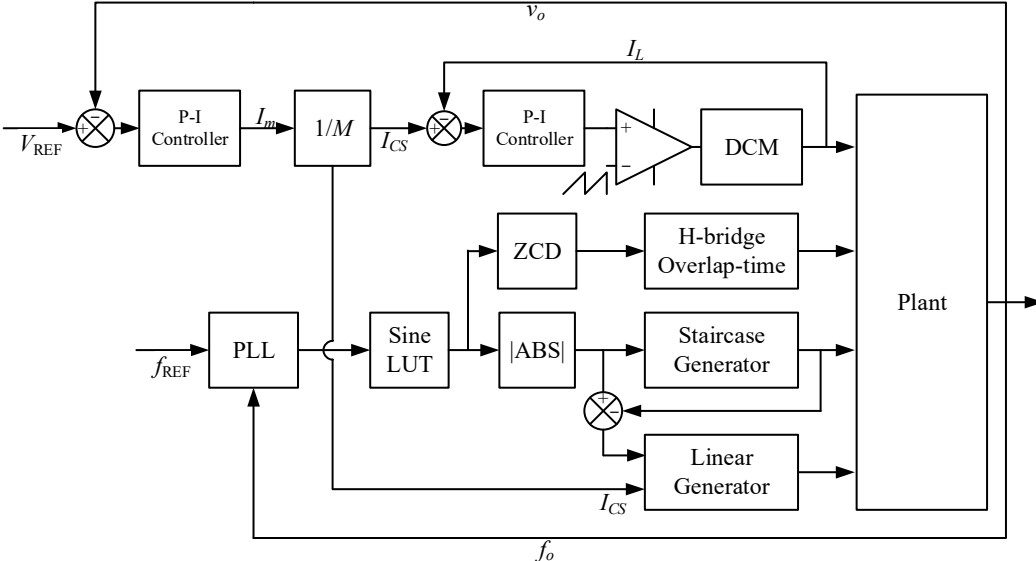

**Figure 9.** Control system.

The load voltage $v_O$ is subtracted with the voltage reference $V_{REF}$ in order to identify an error. This error is then amplified with proportional-integral (P-I) controller to generate the current reference $I_m$. By using Equation (6), the current limit $I_{CS}$ of each current source is determined. In order to control inductor current of DCM $I_L$, the output of current P-I controller is modulated by high frequency sawtooth carrier waveform. In the output frequency control section, the phase output of the PLL drives the sinusoidal look-up table (LUT) to generate sinusoidal waveform reference for H-bridge polarity inversion, staircase current generator, and linear current generator. H-bridge changes its polarity when the zero-crossing detector (ZCD) detects a zero-cross of sinusoidal reference. During overlap time, all switches of H-bridge are turned on to provide free-wheeling current path. The staircase and linear generators have already been explained in the previous section.

### 2.7. Power Losses' Theoretical Analysis

This power losses' theoretical analysis suggests to predict the total dissipated power of the proposed hybrid MCSI that corresponds to the number of levels. The power losses mainly consist of conduction, switching, and gate charge losses [31]. The conduction losses are obtained from modelling the hybrid MCSI circuit. The switching and gate charge losses are calculated from each switching device that is utilized in the proposed system.

The circuit model in Figure 10 represents the proposed hybrid MCSI circuit that is shown in Figure 4. In order to maintain the load voltage $V_L$, the load $R_L$ requires a current of $I_m$. If the parasitic resistance of each current source is assumed equal as $R_{DCM}$, then each current source will generate the current that equals Equation (6).

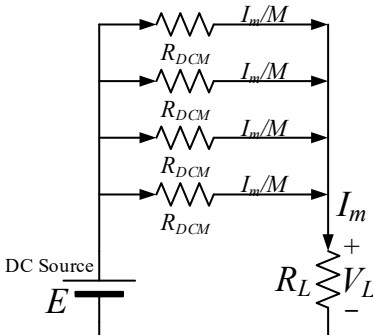

**Figure 10.** Equivalent circuit model of the proposed hybrid MCSI.

Based on Figure 10, the power that dissipated by each current source or DCM is expressed by:

$$P_{DCM} = \left(\frac{I_m}{M}\right)^2 R_{DCM} \tag{18}$$

Thus, the theoretical total conduction losses of the system can be calculated as follows:

$$P_{COND\_TOTAL} = MP_{DCM} = \frac{I_m{}^2}{M} R_{DCM} \tag{19}$$

On the other hand, the switching and gate charge losses are distributed to the H-bridge circuit, the switching module part of DCM circuit, and the buck-chopper part of DCM circuit, respectively. However, the switching losses in the H-bridge are not significant, because the polarity reversal is carried out when the current is near zero ampere and using fundamental switching frequency. Therefore, it can be neglected and only H-bridge gate charge losses are counted. The switching loss of each switching device commonly is expressed by:

$$P_{SW} = E_d f_S \left[ I_d \left( t_r + t_f \right) + C_{OSS} E_d \right] \tag{20}$$

The gate-charge loss of each device commonly is expressed by:

$$P_{GC} = 2Q_g V_g f_S \tag{21}$$

where $E_d$ is the DC link voltage, $f_S$ is the switching frequency, $I_d$ is the passing current through the device, $t_r$ is the rise time of the switching transition, $t_f$ is the fall time of the switching transition, $C_{OSS}$ is the device output capacitance, $Q_g$ is the gate charge, and $V_g$ is the gate voltage. If all calculations of the power losses are plotted into graphics and normalized to the 3-level CSI, the results are shown in Figure 11a–c.

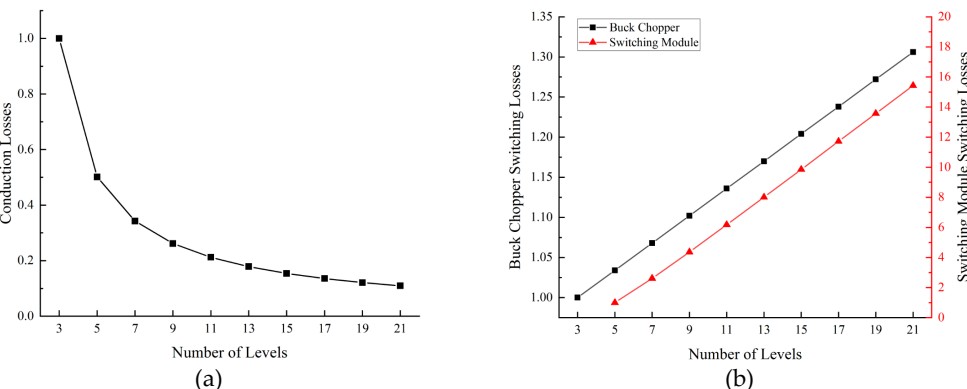

(a)

(b)

**Figure 11.** *Cont.*

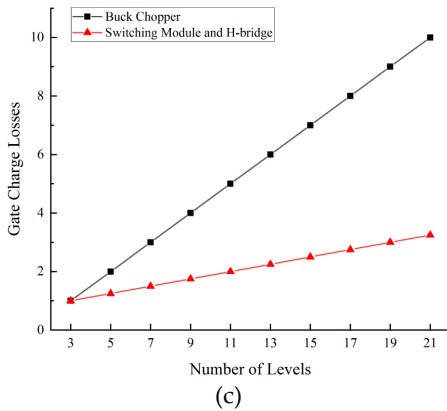

(c)

**Figure 11.** Theoretical MCSI power losses correspond to the number of levels. All data is normalized to 3-level CSI: (**a**) conduction losses; (**b**) switching losses; (**c**) gate charge losses.

Figure 11 shows the correlation between power losses and the number of levels is positive and statistically significant at a high level. Compared to the 3-level CSI, the conduction losses of the system reduced to 10% when implemented on 21-level CSI. In contrast, the switching losses of the buck choppers, the switching losses of the switching module, the gate charge losses of high-switching devices, and the gate charge losses of low-switching devices increased to 130%, 1543%, 1000%, and 325%, respectively. Despite the fact that the conduction losses were reduced significantly, there is a total power losses' analysis limitation due to switching and gate charge losses. Moreover, there is another loss that must be considered, i.e., core loss. Many variables and parameters must be accounted to the analysis in order to investigate the impact of the switching, gate charge, and core losses to the total power efficiency. Therefore, simulation and experimental operations are needed for further power losses investigations.

## 3. Simulation Results

In order to verify the proper operation of the proposed hybrid MCSI and investigate the correlation between efficiency-THD with switching frequencies, output filter capacitor sizes, and the number of levels, several simulations were conducted by using the PSIM application. The first simulation intended to show the operation principle is consistent with the proposed concept. This simulation used parameters that listed in Table 2.

Figure 12a shows the operational simulation result of the proposed hybrid 9-level CSI with class-A linear current compensator in one cycle. Figure 12a consists of the load current, the load voltage, the unfiltered load current, the staircase current, and the linear compensator current. It is shown that the unfiltered load current is the summation between staircase and linear currents. Due to low inductive load, the load voltage seems in phase with the filtered load current.

While Figure 12b shows the fast Fourier transform (FFT) analysis of both the unfiltered current and the load voltage. The horizontal axis shows the harmonic frequency from 0 Hz to 60 kHz, and meanwhile the vertical axis is the normalized harmonic amplitude to the logarithmic unity in percent.

**Table 2.** Parameters for 9-level hybrid multilevel current-source inverters (MCSI) simulation.

| Parameter | Value (Unit) |
| --- | --- |
| Power source DC voltage | 160 V |
| DCM Inductor | 560 µH/90 mΩ |
| Chopper Carrier Frequency | 50 kHz |
| Output Filter Capacitor | 6.8 µH/1 mΩ |
| Load | 10 Ω/1 mH |
| Output AC Voltage/Line Freq | 100 V/60 Hz |

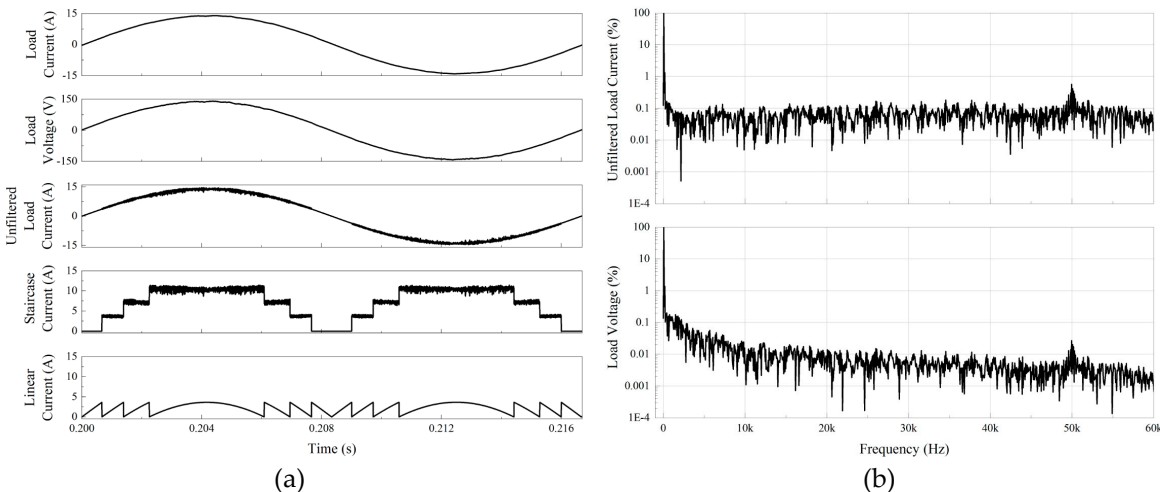

**Figure 12.** The proposed 9-level hybrid MCSI simulation results: (**a**) operation principle; (**b**) fast Fourier transform analysis of the unfiltered load current and the load voltage.

It is shown that the 3rd order of the unfiltered load current harmonic was only around 1%, at 50 kHz switching frequency was around 0.6% and the others were no more than 0.5%. By only using small size output filter capacitor, these higher order harmonics were attenuated to −20dB. As a result, the load voltage harmonic was only around 1%. From this operation simulation result, the generated output voltage THD conforms to the various international standards.

The second simulation intended to explain the correlation between efficiency-THD with switching frequencies, output filter capacitor sizes, and the number of levels as well as their performance comparison with the previous research or conventional method that has been undertaken by author [26]. Figures 13–15 depict the power efficiency and THD-V results of the conventional 9-level system, the proposed 9-level system with class-A linear current compensator, and the proposed 9-level system with class-D linear current compensator, respectively in 3D graphics. The simulation parameters used in this simulation were the same as listed in Table 2, but with various chopper carrier frequencies and output filter capacitors. The various chopper carrier frequencies used in this simulation were 5 kHz (which is denoted as $F_1$), 25 kHz ($F_2$), 50 kHz ($F_3$), 75 kHz ($F_4$), and 100 kHz ($F_5$). While the various output filter capacitor used in this simulation were 680 nF (which is denoted as $C_1$), 2.2 µF ($C_2$), 6.8 µF ($C_3$), 22 µF ($C_4$), and 68 µF ($C_5$).

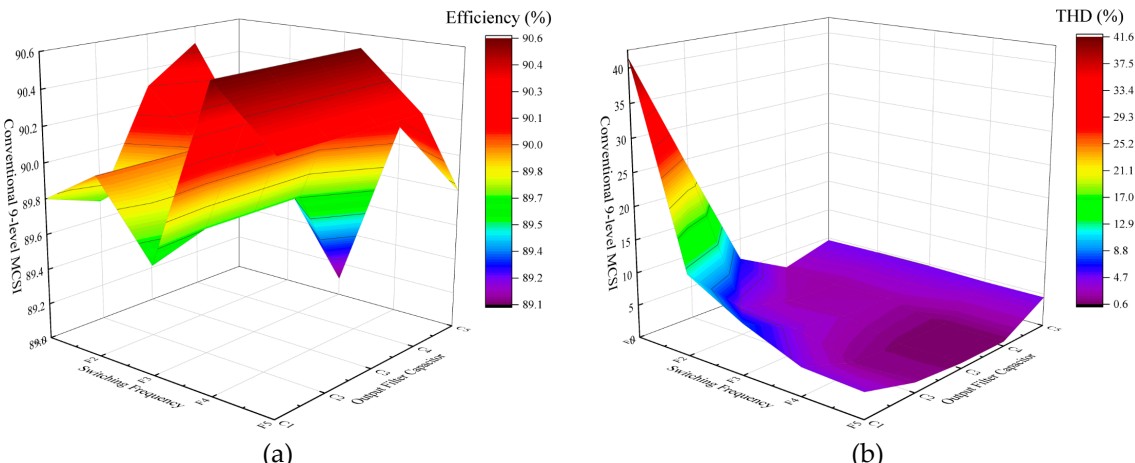

**Figure 13.** The conventional 9-level MCSI performances correspond to the various switching frequencies and output filter capacitors: (**a**) power efficiency; (**b**) voltage total harmonic distortion (THD-V).

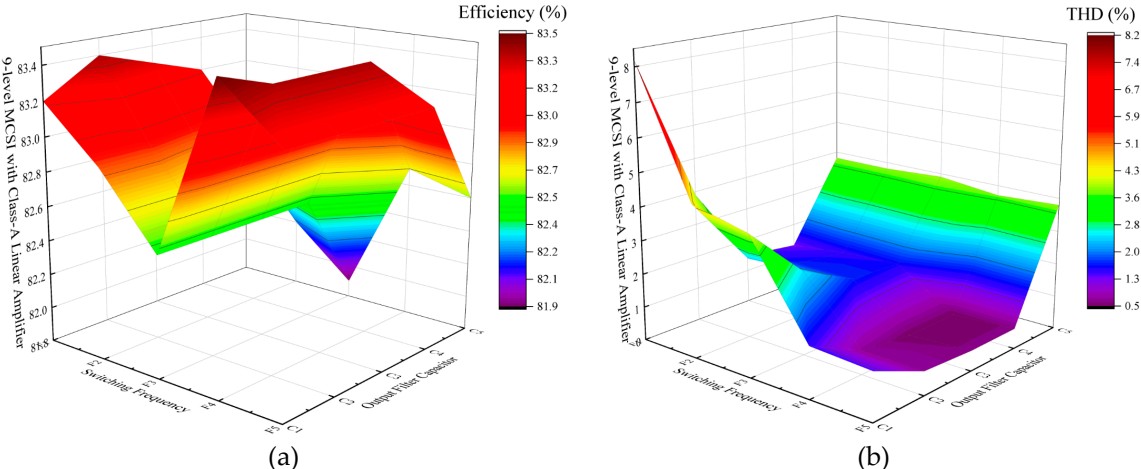

**Figure 14.** The proposed 9-level MCSI with class-A linear current compensator performances correspond to the various switching frequencies and output filter capacitors: (**a**) power efficiency; (**b**) voltage total harmonic distortion (THD-V).

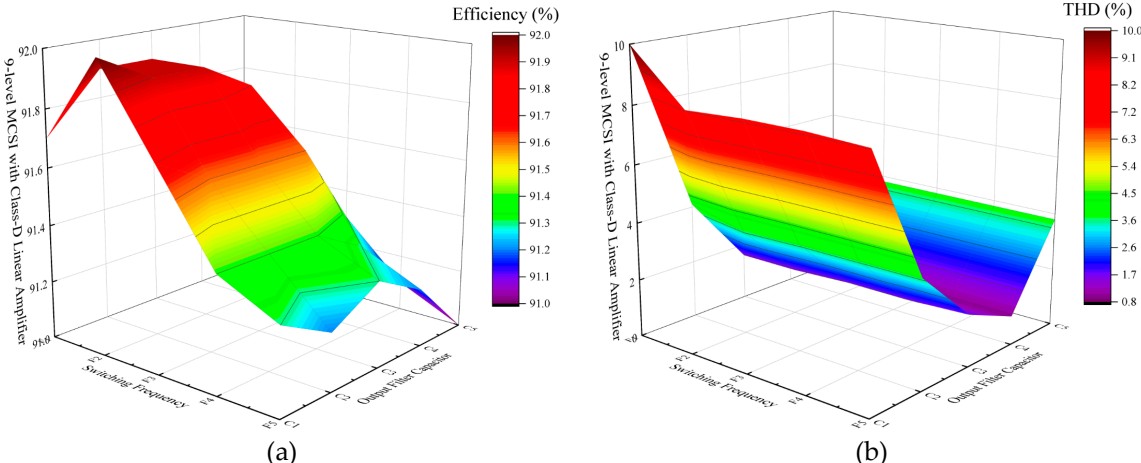

**Figure 15.** The proposed 9-level MCSI with class-D linear current compensator performances correspond to the various switching frequencies and output filter capacitors: (**a**) power efficiency; (**b**) voltage total harmonic distortion (THD-V).

Figure 13 represents the efficiency and THD-V of the conventional method. In Figure 13a, the best efficiency was achieved by the system using 75 kHz switching frequency and 0.68–22 µF filter capacitors. While in Figure 13b, the best THD-V was achieved by the system using 22 µF filter capacitor. Therefore, it can be concluded the optimum performance of the conventional method could be achieved by using 75 kHz switching frequency and 22 µF filter capacitor.

Figure 14 shows the efficiency and THD-V of the proposed method with class-A linear current compensator. In Figure 14a, the best efficiency was given by the system using 75 kHz switching frequency; while in Figure 14b, the best THD-V was given by the system using 22 µF filter capacitor. However, the system used 75 kHz switching frequency and 0.68 µF filter capacitor was chosen as having the best performance, because its THD-V still conforms to the standards while using the smallest capacitor.

Figure 15 depicts the efficiency and THD-V of the proposed method with class-D linear current compensator. In Figure 15a, the best efficiency was given by the system using 25 kHz switching frequency. While in Figure 15b, the best THD-V was given by the system using 22 µF filter capacitor. However, the system used 25 kHz switching frequency and 6.8 µF filter capacitor was selected

as the optimal configuration, because its THD-V still conforms to the standards while using the smaller capacitor.

From analysis of Figures 13–15, it is important to note that despite the lowest efficiency, the class-A linear current compensator greatly reduces the output filter capacitor utilization, while the class-D linear current compensator assists to reduce dependency of using high-switching frequency as well as large capacitor.

## 4. Experimental Results

A laboratory prototype was constructed in order to validate the computer simulation results. For DC current modules, IGBT STGP20V60DF from ST Microelectronics (Geneva, Switzerland) was used as the switches and SiC Schottky diode CREE C3D10060A from Wolfspeed (Durham, NC, USA) was used for the freewheeling and blocking diodes. For the H-bridge circuit, IGBT STGW60V60DF from ST Microelectronics was used as the switches and ultra-fast recovery diode STTH60RQ06 from ST Microelectronics was used as the blocking diodes. FPGA Xilinx Kintex-7 (Xilinx, San Jose, CA, USA) was utilized as the main controller, Yokogawa DL850E ScopeCorder (Yokogawa Measurement, Tokyo, Japan) was used for the oscilloscope, and Yokogawa WT1800 was used as the power analyzer. The complete hardware experimental setup is shown in Figure 16.

The experimental setup was divided into two categories. The first experiment intended to validate the operation principle. The second one was to compare power efficiency and THD-V between simulation and experimental results that correspond to the number of levels. In this experiment, 50 kHz switching frequency and 6.8 μF output filter capacitor were used by the system.

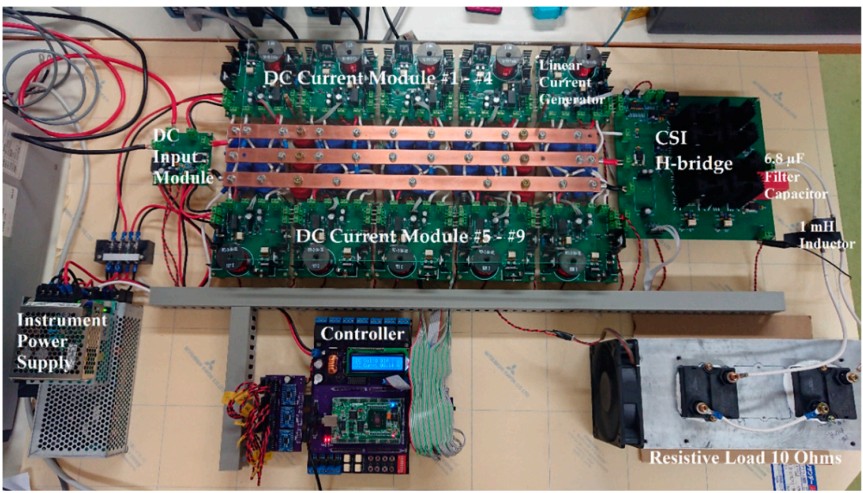

**Figure 16.** Experimental setup.

The experimental setup shown in Figure 16 consists of a DC input module, several DC current modules, a class-D amplifier-based linear current compensator which used similar DC current module, a CSI H-bridge circuit with mounted 6.8 μF output filter capacitor, instrument power supply, and a light inductive load of 10 Ω/1 mH. The DC input module has a function to sense DC voltage and current. Ten DCMs were needed for investigating the impact of the number of levels to the power efficiency with THD-V as well as for the future investigation. Therefore, this experimental setup was designed to be as flexible as possible.

It is plausible that a number of limitations might have influenced the results obtained. Figure 17 shows the experimental results of the proposed 9-level system using class-D linear current compensator. Figure 17a shows the output waveforms and it is shown the load voltage had massive harmonics. The performance was rather disappointing. This was probably as a result of the staircase current ripples existence and the low-switching frequency application for the class-D linear compensator. It is

shown the staircase current ripples were still relatively high even though the interleaved synchronous technique was applied to all DCMs. While it was being measured by power analyzer, the THD-V indicated more than 3%. However, overall it has been confirmed that this proposed technique is possible to realize.

Figure 17b shows the FFT analysis of the AC outputs. Prior to filtering process, the amplitude of each harmonic higher than 100th order reached up to 20%. After being filtered, the majority of harmonics were successfully eliminated. However, the first 100 orders were only attenuated by −6 dB.

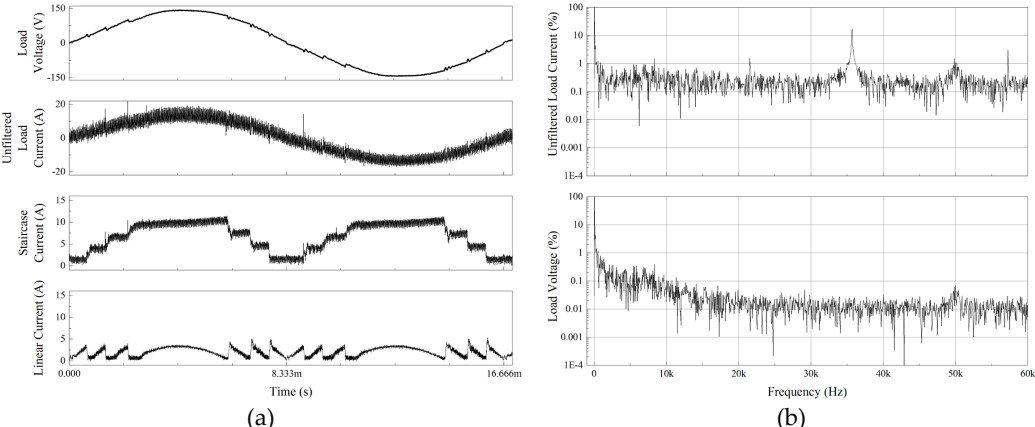

**Figure 17.** Nine-level hybrid MCSI with class-D linear compensator experimental results: (**a**) output waveforms; (**b**) fast Fourier transform analysis of the unfiltered load current and the load voltage.

On the other hand, a class-A linear compensator gave a promising THD-V result. Figure 18a shows that in spite of heavy distorted staircase current waveform, the load current and voltage were successfully compensated. The measured THD-V was only 1.9%. In addition, this result was legitimated by the FFT analysis in Figure 18b. Before being filtered, the load current was indicating that its first 100 harmonic orders exceeded no more than 1%. As a result, the first 100 orders of the load voltage harmonic were successfully damped. However, this information did not include the power efficiency. In order to evaluate the power efficiency of each proposed hybrid MCSI, the second experiment was conducted.

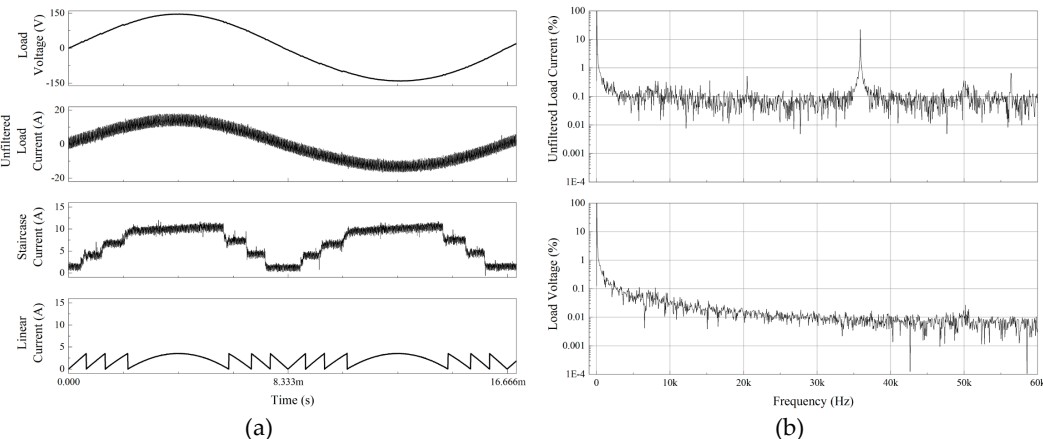

**Figure 18.** Nine-level hybrid MCSI with class-A linear compensator experimental results: (**a**) output waveforms; (**b**) fast Fourier transform analysis of the unfiltered load current and the load voltage.

Figure 19 represents the measured power efficiency along with THD-V that corresponds to the number of levels. Figure 19a is the power efficiency comparison between simulation and experimental

results, while Figure 19b shows the THD-V comparison between simulation and experimental results as well.

In Figure 19a, a slight efficiency uptrend was shown by class-A linear utilization. On the other hand, an efficiency downtrend was shown by class-D linear utilization. However, it had reached the peak of efficiency at 95%. The downtrend was probably caused by insignificant conduction losses reduction at high level, but the other losses such as switching, gate charge, and other losses were significantly increased. A different characteristic was demonstrated by class-A linear utilization which has one less switching based current source. It tends still increasing, but still difficult to exceed 90% due to linear circuit usage. Both simulation and experimental results were shown to be consistent.

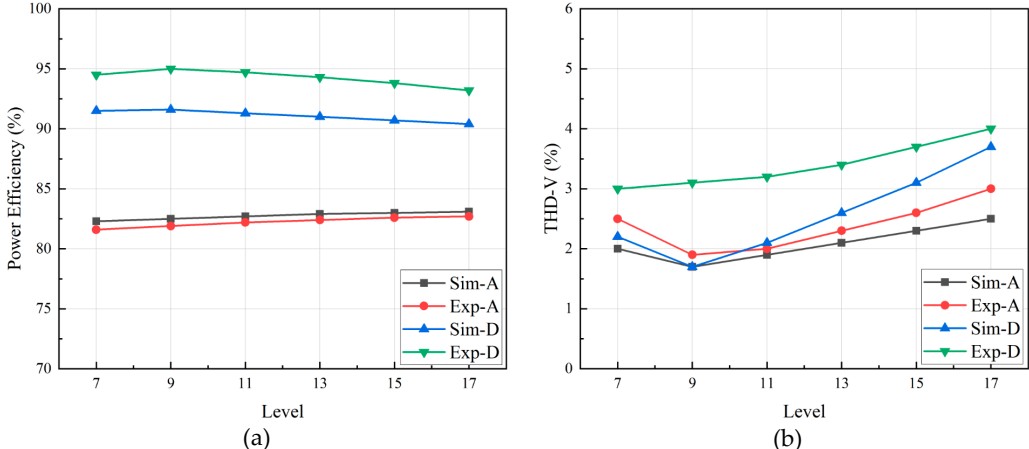

**Figure 19.** Performance comparison between simulation vs experimental results corresponds to the number of levels: (**a**) power efficiency; (**b**) total harmonic distortion of the load voltage.

Figure 19b shows the high-level implementation seemed not giving better THD-V. Unexpectedly, for a higher number of levels, the THD-V was worsening. As explained beforehand, this was caused by the asymmetrical staircase current waveform as the manifestation of the ripple currents from each current source. However, the best result was significant exclusively at 9 levels.

The correlation between the number of levels and power efficiency is worth mentioning because it is understandable in a high number of levels that the reduction of conduction losses becomes meaningless. The most surprising correlation is with the THD-V. The optimal result for the system employing 50 kHz switching frequency and 6.8 μF filter capacitor was obtained at 9-level configuration for both linear amplifier types utilization. Due to tolerable THD-V result, the proposed 9-level hybrid MCSI with class-D linear current compensator was elected as the best configuration. Our technique shows a clear advantage over the conventional method which was employing high-switching frequency for H-bridge circuit. The evidence from this study suggests that class-D linear compensator utilization is preferably applied due to high efficiency. Nevertheless, class-A linear compensator utilization is still essential for power quality application. The current findings add substantially to our understanding of the implementation of each type of linear amplifier in power electronics.

## 5. Conclusions

A novel method for constructing a single-phase multilevel current-source inverter (MCSI) which generates pure sinusoidal output with less switching losses and reduced output filter size has been presented with confirmation from both simulation and experimental results.

In this paper, considerable progress has been made towards the pure sinusoidal output being generated by applying linear amplifier in the MCSI system. There are two types of linear amplifier suitable for this proposed hybrid MCSI, class-A and class-D amplifiers. However, each type has its own characteristics. The class-A linear amplifier has the most linear but challenging power efficiency.

On the other hand, class-D linear amplifier has the best efficiency but tolerable harmonics. According to both simulation and experimental results, the optimal configuration was presented by combining 7-level MCSI with class-D linear amplifier to obtain a 9-level hybrid MCSI.

Taken together, these findings highlight a role for the linear amplifier. In order to construct power quality tools such as active filters or reactive power compensators, class-A linear amplifier utilization is suitable for these applications. On the other hand, a single-phase hybrid MCSI for distributed power generation applications such as photovoltaic inverters or battery-charging systems, requires class-D linear amplifier as the current compensator.

Notwithstanding low power efficiency, class-A linear utilization has presented the best power quality. It will be important that future research investigates the power efficiency improvement of the hybrid MCSI featuring a class-A linear amplifier. Research into solving this efficiency problem is already underway.

**Author Contributions:** Conceptualization, E.R.P. and T.N.; Methodology, E.R.P. and T.N.; Investigation, E.R.P.; Writing—original draft preparation, E.R.P.; Writing—review and editing, E.R.P. and T.N.; Supervision, T.N.

**Funding:** This research was funded by Noguchi Laboratory of Shizuoka University with Research and Innovation in Science and Technology Project (RISET-PRO), Ministry of Research, Technology, and Higher Education of Indonesia.

**Acknowledgments:** Eka Rakhman Priandana was supported by Noguchi Laboratory of Shizuoka University and by Research and Innovation in Science and Technology Project (RISET-PRO), Ministry of Research, Technology, and Higher Education of Indonesia. Any opinions, findings, and conclusions expressed in this material are those of the authors, and do not necessarily reflect the views of the funding agencies. Authors also would like to gratitude anonymous reviewers for their very helpful and constructive comments, which improved this manuscript from the original.

**Conflicts of Interest:** The Authors declare no conflict of interest.

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
