# Peer review of "Pure Sinusoidal Output Single-Phase Current-Source Inverter with Minimized Switching Losses and Reduced Output Filter Size"

_electronics, doi:10.3390/electronics8121556_

Round 1
Reviewer 1 Report
Authors propose a novel method for constructing a single-phase multilevel current-source inverter which generates pure sinusoidal output using a staircase multilevel converter and, to fill the difference to the sinusoidal output, using linear amplifiers.
I agree with the idea, but only if conduction losses are acceptable. This is the hardest point to support this research line, because switched converters just remove conductions losses alternating periods of no voltage or no current. But as long as authors choose a current fed inverter, larger looses of simultaneous voltage and current of linear amplifiers are confined to one branch of a lot of parallel branches.
I think it is a good starting point for discussion.
Author Response
Thank you very much for your comments. We highly appreciated it.
We agree the linear amplifier has larger losses and, as shown in our proposed method, it is confined into one branch of DC link. That is our challenge to minimize the conduction losses, not only in the buck chopper-based current sources, but also in the linear amplifier. Technically, by enlarging the number of levels, the required DC link current is imposed to several branches. Therefore, the constant current sources as well as the linear current amplifier are required to generate small current. As a result, each branch only produces small conduction loss and the total conduction losses of the system will be reduced as well.
Respectfully yours,
EKA

Reviewer 2 Report
The paper proposes the multilevel current source inverter with minimized switching losses and reduced output filter size. The comments are below:
After Eq. (5): “where Im is also known as maximum DC link current”. is Im DC link current or peak amplitude of AC current? (7), the author need to define Isq_avg. Section 2.7: the power losses analysis only includes the device losses which is very small part in the loss distribution. Since the converter is employed several DC current modules with different output current, the inductance of every module is different and causes different loss. please make a complete loss analysis in this section. In addition, this configuration can reduce the switching loss but it dramatically increases the conduction loss. the authors are required to compare the device loss with the conventional configuration as a Table. 11, what is the value of the switching frequency? Also in Fig. 11, what is the MOSFET used to get the Rds,on? What is the current strss on these MOSFET to get the conduction loss? Section 2.7, the device losses included the devices of DC current modules, H-bridge, linear DC current generator, these modules has different switching frequency (fundamental frequency and switching frequency), have the authors considered about it? In simulation section, how many DC current modules is employed? How to design the output current of these modules, are the inductance of these module the same at 560uH? The authors claimed: “ unexpectedly, for higher number of levels, the THD-V was worsening. Probably it was caused by the accumulation of the ripple currents from each current source.” Is it correct?Author Response
Dear Sir,
Please kindly see the attachment. Many thanks in advance.
Respectfully yours,
EKA

Round 2
Reviewer 2 Report
The authors have revised the manuscript well according to the comments. However, there are some minor issues need to be considered:
The authors are required to describe all the parameter, such as: Istaircase in Eq. (14). (it can be added in the sentence: By generalizing Eq. (12) to Eq. (13), the average value of the staircase current (Istaircase) waveform...) In the loss analysis section, the authors have considered about the total conduction loss, switching loss and the gate charge loss. It is correct. However, since many DC- current modules are employed with the inductor at 50kHz, the authors need to consider about the core loss since it is significant in the system (560uH).Author Response
Dear Sir,
Please kindly see the attachment. Many thanks in advance.
Respectfully yours,
EKA
